

# Ecohydrological Optimality in Northeast China Transect

Zhentao Cong[1,2], Qinshu Li[1,2], Kangle Mo[1,2], Lexin Zhang[1,2]

[1]Department of Hydraulic Engineering, Tsinghua University, Beijing, 100084, China
[2]State Key Laboratory of Hydroscience and Engineering, Beijing, 100084, China

*Correspondence to*: Zhentao Cong (congzht@tsinghua.edu.cn)

**Abstract.** Northeast China Transect (NECT) is one of International Geosphere-Biosphere Program (IGBP) terrestrial transects., where there is a significant precipitation gradient from east to west, as well as a vegetation transition of forest-grasslands-dessert. It is interesting to understand vegetation distribution and dynamics under water limitation in this transect. We take canopy cover ($M$), derived from Normalized Difference Vegetation Index (NDVI), as an index to describe the properties of
vegetation distribution and dynamics in NECT. In Eagleson's ecohydrological optimality theory, the optimal canopy cover ($M^*$) is determined by the trade-off of water supply depending on water balance and water demand depending on canopy transpiration. We apply Eagleson's ecohydrological optimality method in NECT based on data from 2000 to 2013 to get $M^*$, then compare with $M$ from NDVI, furthermore to discuss the sensitivity of $M^*$ to vegetation properties and climate factors. The result indicates that the average $M^*$ fits the actual $M$ well (for forest, $M^* = 0.822$ while $M = 0.826$; for grassland, $M^* =$
$0.353$ while $M = 0.352$; the correlation coefficient between $M$ and $M^*$ is 0.81). The result of water balance also matches the field-measured data in references. The sensitivity analyses show that $M^*$ decreases with the increase of LAI, stem fraction, temperature, while increases with the increase of leaf angle and precipitation amount. The Eagleson's ecohydrological optimality method offers a quantitative way to understand the impacts of climate change to canopy cover quantitatively, and provides guidelines for eco-restoration projects.

**Key Words:** NECT; canopy cover; optimality; ecohydrology, climate change

## 1 Introduction

Transect study plays an important role in understanding the role of the terrestrial biosphere in global change (Koch et al, 1995a) The Global Change and Terrestrial Ecosystems (GCTE) project of International Geosphere-Biosphere Program (IGBP) has chosen fifteen transects along with environmental or land-use gradients, aiming at understanding how these factors influence
terrestrial ecosystem and the interaction between biosphere and atmosphere (Koch et al, 1995; Canadell et al, 2002; Austin and Sala, 2002). Northeast China Transect (NECT) was identified as one of the IGBP transects in 1993, with precipitation/moisture as the main driving climate factor (Ni and Zhang, 2000; Zhang and Zhou, 2011). Along with the moisture gradient, the vegetation types vary gradually from forests in the east, to the cropland in the middle, and grassland and bare soil in the west.



Vegetation plays an important role in terrestrial ecosystems. It strongly influences the exchange of energy, substances and moisture between land and atmosphere through photosynthesis, respiration and transpiration (Graetz, 1991; Mcpherson, 2007). At the same time, the vegetation growth condition is largely effected by climate factors, such as precipitation, air temperature and greenhouse gases (Füssler and Gassmann, 2000; Lotsch et al, 2003; Liu and Notaro, 2005). Vegetation is considered as

the indicator of climate, therefore, the study of vegetation growth and distribution is of great importance to cognize the ecosystem construction and functions.

The most common index to describe vegetation performance include Normalized Difference Vegetation Index (NDVI) and vegetation canopy cover. NDVI is a linear combination of remotely sensed near-infrared reflectance and red reflectance. It is an index reflecting the greenness of vegetation canopy and photosynthetic activity (Dorman et al, 2013; Fontana et al, 2008;

Hmimina et al, 2013). Vegetation canopy cover is defined as the fraction of total ground surface covered by vegetation. Semi-empirical relationships between NDVI and canopy cover were used to derive the possible arithmetic expression of canopy cover (Baret et al, 1995; Carlson and Ripley, 1997; Gutman and Ignatov, 1998; Jiang et al, 2006). With the rising attention of the climate change issue, researches about the relationship between canopy cover and climate factors have been conducted in different regions of the world (Zhou et al, 2001; Schultz and Halpert, 1993; Piao et al, 2011; Park and Sohn, 2010; Li et al,

2002; Wang et al, 2003). Nie et al (2011) used correlation analysis to check the relationship between NDVI and climate factors in NECT, with regression equations given for different time scales. Other studies suggest both precipitation and temperature have significant effects on the vegetation along NECT (Piao et al, 2006; Duan et al, 2011; Mao et al, 2012; Peng et al, 2012; Yuan et al, 2015).

Although the statistical models have been established to describe the response of vegetation to climate factors, they cannot

express the underlying mechanism of the response quantitatively. Vegetation models were developed to detect how vegetation reacts to climate change based on the biophysical and physiological processes, including plant life cycle, carbon and nitrogen cycles, but too much data and parameters were required (Myoung et al, 2011). It is a big challenge to build a simplified model that can describe the mechanism of vegetation response to climate change with relatively few parameters. Fortunately, Eagleson (2002) present us a smart theory and method, ecohydrological optimality (Eagleson, 1978a, b, c, d, e, f, g, 1982;

Eagleson and Tellers, 1982). In Eagleson's ecohydrological optimality theory, the vegetation characteristics, such as leaf angle, leaf area index and canopy cover, are determined by the light condition, energy condition, water condition and soil condition in long term average state. Despite the Eagleson's work is regarded as the basis for ecohydrology and of great importance (Hotton et al, 1997; Kerkhoff et al, 2004), limited researches have been conducted using the theory in the practical (Shao et al, 2011; Mo et al, 2015), which partly due to the limitation of long term average state, partly due to the difficulty to measure

vegetation characteristics.

In Eagleson's ecohydrological optimality theory, the optimal canopy cover ($M^*$) is determined by the trade-off of water supply depending on water balance and water demand depending on canopy transpiration. Mo (2015) apply this method in Korqin Sand just for one kind of vegetation. In this study, we apply Eagleson's ecohydrological optimality method in NECT based on



data from 2000 to 2013 to get $M^*$ then compare with $M$ by NDVI, furthermore to discuss the sensitivity of $M^*$ to vegetation properties and climate factors.

## 2 Study area and data

### 2.1 Study area

The Northeast China Transect (NECT) is one of the mid-latitude IGBP terrestrial transects. It ranges from 42° to 46°N and from 106° to 134°E. The major global change gradient is precipitation, which decreases gradually from the eastern mountainous region to the middle farmland and western steppes (Fig. 1). In the east, the annual precipitation is over 600mm; meanwhile, in the west, the annual precipitation is under 200mm/year. The land cover types show a significant zonal distribution from east to west: temperate evergreen conifer-deciduous broad leaf mixed forests, deciduous broad leaf forests

and woodlands in the east, shrublands and crop in the middle, grassland and bare soil in the west (Fig. 1). In this study, we just focus on the growing season, from May to September.

### 2.2 Remote sensing data

Monthly NDVI (MOD13A3), yearly Land Cover Types (MCD12Q1) and 8-day LAI (MCD15A2) datasets derived from Moderate-resolution Imaging Spectroradiometer (MODIS) aboard the Aqua and Terra satellite are applied. These satellite data

are available on NASA website (http://reverb.echo.nasa.gov/).

The spatial resolution of NDVI, Land Cover Types and LAI dataset are 1km, 500m and 1km, respectively. Considering the wide longitudinal and latitudinal extends of NECT, these remote-sensed data are resampled to be 10km x 10km. The MODIS Reprojection Tool (MRT) is applied to define coordinate systems for the images. MRT was also used to generate NDVI and LAI data for growing season of each year.

Canopy cover is defined as the fraction of total ground surface covered by vegetation (Eagleson, 2002). Usually a linear transformation of remote-sensed NDVI is used to calculate actual canopy cover ($M$) (Gutman and Ignatov, 1998; Jiang et al, 2006):

$$M = \frac{NDVI - NDVI_{min}}{NDVI_{max} - NDVI_{min}} \qquad (1)$$

in which $NDVI_{min}$ is the NDVI of barren soil, and $NDVI_{max}$ is the NDVI of forests. Though the land cover types did not change

much from 2000 to 2013, it was hard to define the real barren soil or the forest areas, as the satellite data is not accurate in some respects. We considered the area sensed as barren soil for every year is the barren soil area, and the $NDVI_{min}$ is the spatial average of barren area NDVI. Similarly, the area sensed as forests every year is considered as forests area, and the spatial average of forests NDVI is $NDVI_{max}$. In this study, $NDVI_{min}$ and $NDVI_{max}$ are 0.05 and 0.63 respectively, which means the canopy cover can be regarded as 1 if the NDVI is above 0.63 and as 0 if the NDVI is below 0.05.





## 2.3 Meteorological data

The meteorological data used in this study from 2000 to 2013 is provided by China Meteorological Data Sharing Service System (http://cdc.cma.gov.cn). The spatial distribution of the 45 meteorological stations is shown in Fig. 1. Atmospheric pressure ($P_a$), wind speed ($W_{nd}$), average air temperature ($T_a$), net radiation (Rn) (estimated by sunshine hours ($S_h$) and air

temperature (Allen, 1998)), relative humidity ($R_h$), minimum air temperature ($T_n$), and maximum air temperature ($T_m$) are required to calculate the potential evapotranspiration by Penman Monteith Equation (Ni and Zhang, 2000; Eagleson, 2002). Kriging interpolation method is applied to generate the spatial distribution of the meteorological factors and potential evapotranspiration. The spatial resolution is 10 km to be consistent with that of remote sensing data.

## 3 Methodology

Eagleson proposed three hypotheses in his ecohydrological optimality theory. He considered the climate and vegetation can influence and adapt to each other on different time scales. First, when climate and soil changes in short time period, the canopy cover will adjust its value to maximize the soil moisture. Second, as the time scales get longer, the species whose potential transpiration efficiency make the soil moisture highest will be selected through natural selection. Third, the soil properties will be altered to ensure the species get their maximum canopy cover (Eagleson, 2002; Hatton et al, 1997). These hypotheses

mentioned two important canopy state variables, i.e., the canopy cover ($M$) and canopy conductance ($k_v$). Canopy conductance is defined as the ratio of potential rates of transpiration $E_v$ and soil surface evaporation $E_{ps}$ (Eagleson, 1978d; Eagleson, 2002):

$$k_v = \frac{E_v}{E_{ps}} \tag{2}$$

The potential evapotranspiration $E_{ps}$ is calculated by Penman Equation:

$$\lambda E_{ps} = \frac{\Delta R_n + \rho c_p [e_s(T) - e]/r_a}{\Delta + \gamma_0} \tag{3}$$

The canopy transpiration rate $E_v$ is calculated by Penman Monteith Equation:

$$\lambda E_v = \frac{\Delta R_n + \rho c_p [e_s(T) - e]/r_a}{\Delta + \gamma_0 (1 + r_c/r_a)} \tag{4}$$

where

|        |                                                                            |
|--------|----------------------------------------------------------------------------|
| $E_{ps}$ | potential rate of evaporation from a wet, simple surface, mm/day |
| $E_v$  | rate of canopy transpiration, mm/day |
| $\lambda$ | latent heat of vaporization of water = 597.3 cal/g (at 0 ∘C) |
| $\Delta$ | the slope of the saturation vapor pressure vs. temperature curve, $P_a$/K; |
| $R_n$  | net solar radiation, cal/(mm$^2$.day); |
| $\rho$ | fluid mass density, g/mm$^3$; |
| $c_p$  | specific heat of air at constant pressure, cal/(g·K); |
| $e_s(T)$ | saturation vapor pressure at the temperature of the evaporation site, Pa; |
| $e$    | partial pressure of water vapor, Pa. |
| $r_a$  | the lumped atmospheric resistance over the 2m above the canopy top, day/mm; |




$r_c$       the lumped resistance to flow through the canopy which does not vary with water supply, day/mm;

$\gamma_o$       the surface psychrometric constant, Pa/K.

When the stomates fully open, the canopy transpiration rate $E_v$ will reach its maximum value – potential canopy transpiration $E_{pv}$, thus making $k_v$ to be its maximum value as well, which is called the potential canopy conductance $k_v^*$:

$$k_v^* = \frac{E_{pv}}{E_{ps}} = \frac{1 + \Delta/\gamma_0}{1 + \Delta/\gamma_0 + (1-M)\left(\frac{r_c}{r_a}\right)_{M \to 0} + M\left(\frac{r_c}{r_a}\right)_{M=1}} \tag{5}$$

where $(r_c/r_a)_{M \to 0}$ is the resistance ratio for open canopies, which is related to the exponent relating shear stress on foliage to horizontal wind velocity and horizontal leaf area index. $(r_c/r_a)_{M=1}$ is the resistance ratio for closed ($M=1$) canopies whose mainly influence factor is the ratio of stem height $h_s$ and tree height $h$. According to the Eq. (5) and explanations above, the resistance ratio can be fixed once the vegetation specie is given. The potential canopy conductance $k_v^*$ is inversely proportion to the canopy cover $M$. The $k_v^*$-$M$ curve is called water demand curve.

The relationship between $k_v^*$ and $M$ can be also described by water balance equation. In growing season, the average inflows and outflows of the soil column:

$$P_\tau - m_v E[E_r] - \Delta S = m_v E[R_{sj}] + E[E_{T\tau}] + m_\tau v - m_\tau w \tag{6}$$

where

$P_\tau$       growing season precipitation, mm;

$m_v$       the number of independent storm times, dimensionless;

$E_r$       the storm surface retention depth, mm

$R_{sj}$       storm rainfall excess, mm

$\Delta S$       average carryover (from dormant season to growing season) soil moisture storage, mm;

$m_\tau$       the growing season length, day;

$v$       percolation to water table, mm/day;

$w$       capillary rise from water table, mm/day;

E[ ]       means the expected value of [ ].

Some assumptions are made to describe each item (Eagleson 1978a, b, c, d, e, f, g). Thus, the water balance of the growing season can be expressed as:

$$M k_v^* = \frac{V_e}{m_{tb} E_{ps}} \tag{7}$$

where $V_e$ is the volume of soil moisture (per unit of surface area) available for exchange with atmosphere during average interstorm period (mm), $m_{tb}$ is mean time between storms (h), more detail can be found in the appendix.

Equation (7) describes water supply in naturally selected canopy moisture state, while Eq. (5) describes water demand of fixed vegetation species. By drawing these two lines in a figure (Fig. 2 ), we can notice that the water demand grows with the increase of $M$, but as water is limited, water supply decreases under $M$ enhancement. The intersection point of these two lines is the theoretical optimal canopy cover and potential canopy



conductance in the vegetation state-space. This method is applied in each grid (10km x 10km) of NECT area, and the input values are listed in Table 1.

## 4 Results and discussion

### 4.1 Canopy cover of NECT

The observed canopy cover $M$ shows a significant gradient ranging from 1 in the east forests to 0 in the west desert (Fig. 3a). The dark blue area is mainly the forests of Changbai Mountains, where the average canopy cover reaches up to 0.83. The light blue area is the Songnen Plain. Songnen Plain is one of the most famous commodity grain bases, rich in corn, sorghum, soybean, wheat and paddies (Zhou and Wang, 2003), with the average canopy cover of 0.55. Farther westward, there is Horqin Sand, in which most of the vegetation is grass. Then there is a narrow northeast-southwest-oriented band with relatively higher value

(blue color) at around 120°E, which is mainly caused by the elevation. The band is the location of Greater Khingan Mountains. The east slope of the Greater Khingan Range is very steep, thus the maritime monsoon can bring a lot of rainfall, causing the existence of forest ecosystem. However, most of the vegetation on the west slope is grass, mainly because of the gentle gradient and dry climate (Guo and Zhang, 2013). The grassland is Inner Mongolia steppe.

The Ecohydrological optimality theory is applied in this study to simulate theoretic optimal canopy cover ($M^*$) of NECT. As

shown in Fig. 3b, the modeled canopy cover has the same trend with the actual $M$ but transits more smoothly, which is mainly caused by the interpolation of meteorological data. The blank grids in the simulation result are due to the missing data of $LAI$. The result indicates that the average $M^*$ fits the average $M$ well (for forest, $M^* = 0.822$ while $M = 0.826$; for grassland, $M^*$ =0.353 while $M = 0.352$; the correlation coefficient between $M$ and $M^*$ is 0.81). The corresponding areas are highlighted in the figure of spatial distribution of $\Delta M$, defined as $M^*$ minus $M$ (Fig. 3c). The spatial average $\Delta M$ is only -0.050 for the whole

NECT area, meanwhile, there are 45.7% pixels of NECT area where $\Delta M$ value is between -0.1 to 0.1. There are three regions where the differences between $M$ and $M^*$ is relatively large. Region 1 is Yanbian Korean Autonomous Prefecture. The simulation result is relatively small mainly because of human activity. The Natural Forest Protection Project (NFPP) has been conducted in Northeastern China since 1998, aiming at protecting the natural forest resources (Wei et al., 2014). Yanbian forest acreage has increased by 800 km² during the first stage of NFPP. The dark red area is Hunchun City. Hunchun is a

representative nature reserve, and the forest acreage has increased by 9,009 ha during 1999 to 2012 (Li, 2014). Region 2 is the southern Xilin Gol Grassland. In the past decades, Xilin Gol Grassland is extremely dry and had been suffering from severe degradation (Tong et al, 2002). The Beijing-Tianjin Sand Source Control Project is undertaken to improve the canopy cover of degraded grassland. Over 66,000 water source projects and 47,000 water saving irrigation projects increased the water supply of this area, thus contributing to the increase of vegetation activities (Yu et al, 2010). The irrigation part is not considered

in the Eagleson's water balance system, which leads to the deviation of the modeled results. In the crop region (the blue frame in Fig. 3c), some $M^*$ are higher than $M$ while some are lower. This is because of the close relationship between canopy cover and crop growth stage. The growth process of various crops are different, and the timing of plantation and harvesting are



mainly effected by human intervention rather than natural processes (Liu et al, 2013; Kim and Wang, 2005). Meanwhile, the water supply for the crop is not only from natural hydrological cycle but also from agricultural irrigation, which is not considered in the theory.

The correlation coefficient $R$ between $M$ and $M^*$ is 0.81, which indicates the Ecohydrological Optimality theory is applied well in NECT during long-term period. Previous researches suggest there are lagged relationship between NDVI and climate factors, and the time lags are different at different region scales or different biomes (Braswell et al., 1997; Piao et al., 2003; Li et al., 2011; Hu et al., 2011; Bao et al., 2015). Fig. 4 shows that in grassland area, the variation amplitude of $M$ is smaller than $M^*$, the delay usually happens within a year; while in forest area, there is a trend delay across the years. For example, the $M^*$ is increasing from 2007 to 2009, but the increasing trend of $M$ does not appear until 2009 to 2010. This can be explained by the vegetation adaptation strategy to climate changes. Eagleson's Theory describes how vegetation adapt to climate change in a relatively long term (Eagleson, 2002). Once climate changes, it takes years for vegetation to reach its optimum canopy cover.

## 4.2 Water balance components

As NECT is spanning a wide range from west to east, and the vegetation and climate vary significantly, the NECT is divided into three parts according to land cover types: forests, cropland and grassland (Ni and Zhang, 2000). The proportions of the water balance components for annual average growing season are calculated for each part, as shown in Table 2. According to the researches conducted before, in grassland area, the interception was 20.86% and 7.88% for shrub and grass respectively (Peng et al, 2014), and the runoff of Xilin Gol grassland occupies around 0.046%~1.8% (Wang, 2008; Miao, 2008). In forest area, the dominant tree species are Pinus koraiensis (Pk), Quercus mongolica (Qm), Populus davidiana (Pd) and B. platyphylla (Bp) (Chen, 2001; Zhang and Zhou, 2009). The interception consists 19.61% for Pk and 14.97% for Bp in Great Greater Khingan Mountains, and 10.20% for Pk in Changbai Mountains (Cai et al, 2006; Wang et al, 2006). The runoff coefficient of Suifen River and Secondary Songhua River are around 20%~30%, both of which are located in forest area (Huang, 1999; Song, 2010). The simulated interception and runoff for both grassland and forest area are within the observed range, which demonstrates the reasonability of this theory. The negative value of $\Delta S$ in forest area means a recharge of soil moisture. As the air temperature in the non-growing season is low, most of the precipitation is snow rather than rain, so the water is frozen in the soil and melts in the next spring (Fan et al, 2006; Yang et al, 2006). Therefore, most of the water is stored in the dormant season for the vegetation grow in the next growing season.

The rationality of the calculated proportions of water balance components for each part demonstrates the applicability of the optimality theory. By adapting this method, it is much easier to figure out the allocation of precipitation if the vegetation and soil conditions are known.

## 4.3 Sensitivity of $M^*$ to vegetation properties

$LAI$, $\beta$ and $h_s/h$ control the physical and biological processes of plant canopies, such as interception and evaporation (Chen and Black, 1992; Asner, 1998; Huete et al. 2002). $\beta$ and $LAI$ are the dominant parameters for the interception calculation, thus





leading to the variance of water supply for vegetation growth. $\beta$ and $h_s/h$ affect the plant evaporation through affecting the resistance ratio, which influences the water demand curve (Eagleson, 2002). The thresholds of the three parameters are from the experiments conducted before (Du, 2004; Wang et al, 2008; Rauner, 1976; Eagleson, 2002). Fig. 5 shows the different reactions of optimal canopy cover to vegetation species change between grassland and forest area. $M^*$ increases with the

increase of leaf angle and decrease of stem fraction and $LAI$. Mo (2015) studied the relationship between vegetation properties and optimal canopy cover in Horqin Sands, China, and got the same conclusion. In grassland area, the water demand curve is more sensitive to the variation of $h_s/h$ compared to $\beta$. $M^*$ decreases by 0.037 as $h_s/h$ increases 0.10. The water supply curve changes a lot with the change of $LAI$, but slightly with $\beta$ or $h_s/h$. In forest area, $M^*$ is less sensitive to $h_s/h$ than $\beta$ or $LAI$. Because the average stem fraction of trees (0.4~0.5) is usually larger than grasslands or shrub (0.0~0.1), the water demand

curve is much gentler (Eagleson, 2002), and changes little with $h_s/h$. The forest interception consists 14.24% of precipitation during growing season, which is larger than that of grassland. $M^*$ increases by 0.108 and 0.094 with the 0.56 decrease of $\beta$ and 2.45 of $LAI$, respectively (Table 3).

The sensitivity of $M^*$ to vegetation properties can be used to offer advices about species choice and plant density to eco-restoration projects. If the purpose is to increase canopy cover, different strategies should be conducted in different area. For

grassland area, shrubby or herbaceous plants with low $h_s/h$ value are more welcome. Nevertheless, in forest area, as $h_s/h$ does not affect canopy cover that much, more considerations should be taken into choosing the species with relatively lower $\beta$ and $LAI$ values. However, vegetation with a larger canopy cover always requires more water to maintain functions (Woodward and Mckee, 1991; Zhang and Zhou, 2011). If the plant species are determined, the optimum canopy cover can be calculated, and the upper limit for plant density can be estimated.

**4.4 Sensitivity of $M^*$ to climate factors**

Studies of relationship between climate factors and vegetation growth condition reveal that precipitation and temperature are the two dominant factors that affect $M^*$ (Ichii et al, 2002; Liu et al, 2015). Under this framework, the variation of precipitation ($P_\tau$) affects the availability of water, thus changing water supply curve; air temperature ($t_a$) affects not only water supply but also water demand, through changing resistance ratio and evaporation (Fig. 6). In grassland area, $M^*$ exhibits a positive

relationship with precipitation but a negative relationship with air temperature (Fig. 6(a)(b)), of which is consistent with studies conducted before (He et al, 2015; Peng et al, 2012). This can be explained by the limited water supply in arid and semi-arid regions, and that the increase of air temperature enhances transpiration and evaporation intensity (Duan et al, 2011; Mao et al, 2012). The variation of grassland $M^*$ during 2000-2013 (Fig. 4(a)) also shows the similar trend of $M^*$ and precipitation, while the trend of air temperature is different from that of precipitation in most years. In forest area, $M^*$ increases with the increase

of precipitation and decrease of air temperature, but the variance of grassland $M^*$ is less than that of forest with the same range of air temperature, which indicates the forest plants are more sensitive to air temperature than grassland. However, the result is different to previous studies. Most correlation analysis of NDVI with air temperature and precipitation shows that in forest area, NDVI increases with the increase of air temperature and decrease of precipitation, because air temperature is the dominant



factor in humid areas, and the light use efficiency increases under elevated air temperatures (Peng et al, 2012; Wang et al, 2014; Liu et al, 2011). The difference may be caused by the deficient hypotheses of the theory. Under this framework, the surface runoff is assumed to be Hortonian, but in most humid areas, the runoff is saturation excess. The improper hypothetical runoff mechanism leads to the deviation of runoff in water-sufficient areas, thus causing the deviation of water supply curve.

During the past few years, a lot of studies were carried out to detect the relationship between climate and vegetation. Nie et al. (2012) applied the wavelet regression analyses in NECT, demonstrating that the most suitable time scales for evaluating the impact of climate factors was 160-day for most stations. Zhang and Zhou (2011) conducted correlation analyses to study the relationships between net primary productivity (NPP) and climate factors, suggesting that precipitation played more important role than air temperature for unchanged biomes, but when the biomes changed, the rate of NPP change had more close relation

to air temperature. Despite variance methods were developed by hydrologists to explore the vegetation response to climate, overwhelming majority of them are statistical. To understand the mechanism of climate influence on vegetation, more and more models with vegetation biophysical and physiological processes are developed. Nevertheless, these vegetation models require too many inputs, and some of the data are hard to get (Myoung et al, 2011). Different from the models above, Eagleson's ecohydrological optimality theory can not only explore the mechanism of canopy cover distribution, mainly from water balance

perspective, but also easy to conduct. The optimality theory provides a new way to explore the quantitative relation between vegetation and climate factors.

## 5 Conclusion

In this study, remote-sensed NDVI is used to generate actual canopy cover of NECT, while the ecohydrological optimality method has been applied to calculate the optimal canopy cover. The proportions of water balance components have been

explored, as well as the influence of vegetation properties and climate factors to optimal canopy cover. The main conclusions are as follows:

(1) The observed canopy cover $M$ shows a significant decreasing gradient from east forests to west. The modeled canopy cover $M*$ has the same trend with $M$ but transits more smoothly, which is mainly caused by the interpolation of meteorological data. The relatively lower $M*$ in Yanbian Korean Autonomous Prefecture and Xilin Gol Grassland is mainly because of human

activity. The correlation coefficient $R$ between $M$ and $M*$ is 0.81, which indicates the Ecohydrological Optimality theory is applied well in NECT during long-term period. There is a two-year-lag between $M*$ and $M$ during 2002-2012, due to the long-term adaptation strategy of vegetation to climate change.

(2) The proportions of the water balance components are calculated for three parts: forest, cropland and grassland. The simulated results are within the observed range, which demonstrates the reasonability of this theory. By adapting this method,

it is much easier to figure out the allocation of precipitation with fixed vegetation and soil conditions.





(3) $M*$ has the positive relationship with $\beta$ and negative relationship with $h_s/h$ and $LAI$. Grassland plants are more sensitive to $h_s/h$ and $LAI$ compared to $\beta$, while forest plants are more sensitive to $\beta$ and $LAI$ than $h_s/h$. The sensitivity of $M*$ to vegetation properties can be used to offer advices about species choice and plant density to eco-restoration projects.

(4) Precipitation and temperature are the two dominant climate factors that affect $M*$. $M*$ increases with the increase of precipitation and decrease of air temperature. Eagleson's ecohydrological optimality theory offers an opportunity to explore the quantitative relation between vegetation and climate factors from the mechanism, but the runoff mechanism description in wet region still needs improvement.

**Acknowledgement**

This work was supported by National Natural Science Foundation (No. 51479088, 41630856).

**Appendix**

**Algorithm of optimal canopy cover**

Eagleson made several assumptions for each item of Eq. (4). Poisson precipitation model was used to simulate the precipitation process by random storm depth and duration (Eagleson, 1978b). The probability density functions of storm depth and storm duration are incomplete gamma and exponential distribution, respectively. The growing season precipitation can be expressed as:

$$P_\tau = m_v/m_h \tag{A.1}$$

where $m_h$ is the mean storm depth, mm.

Surface retention $m_v E_r$ is the water held on the surface during the rainstorm of duration. The total surface retention is proportioned by bare soil and vegetation canopy:

$$E[E_r] = (1 - M)E[E_{rs}] + ME[E_{rv}] \tag{A.2}$$

where $E_{rs}$ and $E_{rv}$ are the surface retention loss of bare soil and vegetation canopy, and can be further expressed as:

$$E[E_r] = (1 + M\eta_0\beta L_t)h_0 = \overline{h_0} \tag{A.3}$$

where $\eta_o$ is the ratio of stomated leaf area to illuminated leaf area, dimensionless; $\beta$ is the cosine of leaf angle, dimensionless; $L_t$ is the foliage area index, dimensionless; $h_o$ is the surface retention depth, mm. The interception depth retained on the horizontal projection of leaves is assumed to be 1.00 cm (Eagleson, 1978d).

Average carryover soil moisture storage $\Delta S$ (mm) is determined by the soil profile and seasonality (Eagleson, 2002):

$$\Delta S = -[P_d - (1 - M)E_{psd}m_d - Y_d] \tag{A.4}$$

where $P_d$ (mm), $E_{psd}$ (mm/day), $m_d$ (day), and $Y_d$ (day) are the precipitation, evaporation, days and runoff in the non-growing season, respectively.





Assume that there is no surface inflow from outside of the region, and the surface runoff is Hortonian (Eagleson, 1978e). When the storm intensity $m_i$ and storm duration $m_{tr}$ are independent random variables, the storm surface runoff $m_v R_{sj}$ is:

$$\mathrm{E}(R_{sj}) = m_h e^{-G - 2\sigma^{3/2}} \tag{A.5}$$

where

$$\mathrm{G} \equiv \omega K(1) \left( \frac{1 + s_0^c}{2} \right) \tag{A.6}$$

$$\sigma \equiv \left[ \frac{5 n_e \lambda_0^2 K(1) \psi(1) (1 - s_0)^2 \phi_i(d, s_0)}{6 \pi \delta m \kappa_0^2} \right] \tag{A.7}$$

where

$s_o$     space-time average soil moisture in the root zone, dimensionless;

$\omega$     $= 1/m_i$;

$K(1)$     effective saturated hydraulic conductivity of soil, cm day$^{-1}$;

$n_e$     effective soil porosity, dimensionless;

$\lambda_0$     scale parameter of probability density function of storm depth, cm$^{-1}$;

$\psi(1)$     saturated matrix potential of soil, cm;

$\phi_i$     sorption diffusivity, dimensionless;

$\delta$     $= 1/m_{tr}$, day$^{-1}$;

$m$     soil pore size distribution index, dimensionless;

$\kappa_o$     shape parameter or distribution index of storm depth, dimensionless.

Evapotranspiration consists of bare soil evaporation and vegetal transpiration:

$$\mathrm{E}[E_{T\tau}] = m_v m_{tb} E_{ps\tau} [(1 - M)\beta_s + M k_v^* \beta_v] \tag{A.8}$$

where $m_{tb}$ is the mean time between storms, day; $E_{ps\tau}$ is the potential free water surface potential evaporation during growing season, mm/day; $\beta_s$ and $\beta_v$ are the bare soil evaporation efficiency and canopy transpiration efficiency respectively (Eagleson, 1978d).

The percolation rate is mainly affected by $s_o$ (Eagleson, 1978f):

$$\mathrm{v}(s_0) = K(1) s_0^c \tag{A.9}$$

The capillary rise is considered to be 0 due to the deep water table in NECT.

Using Eq. (A.1)~(A.9), Eq. (4) gives the water balance of growing season as:

$$1 - e^{-G - 2\sigma^{3/2}} - \frac{\overline{h_0}}{m_h} + \frac{\Delta S}{m_v m_h} = \frac{m_{tb} E_{ps}}{m_h} [(1 - M)\beta_s + M k_v^* \beta_v] + \frac{m_\tau K(1)}{P_\tau} s_0^c \tag{A.10}$$

Eq. (A.10) can be simplified into

$$M k_v^* = \frac{V_e}{m_{tb} E_{ps}} \tag{A.11}$$

where, $V_e = m_h - m_h e^{-G - 2\sigma^{3/2}} - \overline{h_0} + \frac{\Delta S}{m_v} - \frac{m_\tau K(1)}{m_v} s_0^c \tag{A.12}$





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





**Table 1: The terminology, interpretation, units and values of inputs**

| | Terminology | Interpretation and units | Value |
|---|---|---|---|
| Remote Sensing Data | $f_c$ | Average vegetation cover of growing season | 0.00 ~ 1.00 |
| | $M_d$ | Agerage vegetation cover of non-growing season | 0.00 ~ 1.00 |
| | $l_t$ | Leaf Area Index (LAI) in growing season, dimensionless | 0.00 ~ 4.70 |
| | $l_{td}$ | Leaf Area Index (LAI) in dormant season, dimensionless | 0.00 ~ 1.70 |
| | $m_t$ | length of the growing season, days | 153 |
| | $m_d$ | length of the non-growing season, days | 212 |
| | $E_{pst}$ | free water surface potential evaporation during growing season, mm/d | 3.6 ~ 4.4 |
| | $E_{psd}$ | free water surface potential evaporation during dormant season, mm/d | 0.7 ~ 1.0 |
| Meteorological Data | $P_\tau$ | precipitation in growing season, mm | 149.7 ~ 624.3 |
| | $P_d$ | precipitation in dormant season, mm | 26.2 ~ 226.3 |
| | $t_0$ | average temperature in growing season, ℃ | 16.12 ~ 21.24 |
| | $m_{tb}$ | mean time between storms, days | 4.65 ~ 6.35 |
| | $m_{tr}$ | mean storm duration, days | 0.37 ~ 0.64 |
| | $\gamma_0$ | surface psychrometric constant, Pa/K | 0.06 |
| Vegetation Data | $m$ | exponent relating shear stress on foliage to horizontal wind velocity, dimensionless | 0.5 |
| | $n$ | number of sides of each foliage element producing surface resistance to wind, dimensionless | 2 |
| | $\eta_0$ | eta0 = stomated leaf area / illuminated leaf area, dimensionless | 2.50 |
| | $h_0$ | surface retention depth, mm | 1.00 |
| | $\beta$ | cosine of leaf angle, dimensionless | 0.45 |
| Soil Data | $h_s$ | stem height (i.e., height of crown base above substrate), m | |
| | $h$ | height of tree from ground surface to top of crown, m | |
| | $m$ | soil pore size distribution index, dimensionless | 0.50 |
| | $n_e$ | effective soil porosity, dimensionless | 0.45 |
| | $d$ | diffusivity index of soil, dimensionless | 4.30 |
| | $\psi$ | saturated matrix potential of soil, mm | 900.0 |
| | $k$ | effective saturated hydraulic conductivity of soil, mm/d | 29.4 |
| | $s_0$ | space-time average soil moisture concentration in the root zone, dimensionless | 0.30 ~ 0.62 |





**Table 2. Water balance components of different land cover types**

| | | grassland | | cropland | | forests | |
|---|---|---|---|---|---|---|---|
| result | $M$ | 0.352 | | 0.548 | | 0.826 | |
| | $M^*$ | 0.353 | | 0.557 | | 0.822 | |
| | | mm | /P | mm | /P | mm | /P |
| Water balance component | *Precipitation* | 253 | 100.00% | 414 | 100.00% | 478 | 100.00% |
| | *Interception* | 29 | 11.61% | 39 | 9.29% | 68 | 14.24% |
| | *Runoff* | 1 | 0.19% | 119 | 28.77% | 119 | 24.92% |
| | $\Delta S$ | 91 | 36.29% | 14 | 3.34% | -74 | -15.47% |
| | *Evaporation* | 131 | 51.90% | 243 | 58.60% | 365 | 76.31% |

5  **Table 3: The variance of inputs and their corresponding $M^*$**

| inputs | variation range | Grassland $M^*$ | Forest $M^*$ |
|---|---|---|---|
| $\beta$ | 0.01~0.57 | 0.346~0.358 | 0.755~0.863 |
| $LAI$ | 0.10~2.55 | 0.313~0.357 | 0.770~0.864 |
| $h_s/h$ | 0.00~0.10 (grassland); 0.35~0.45(forest) | 0.317~0.354 | 0.817~0.827 |
| $P_t$ | 24.26~26.26 (grassland); 46.83~48.83(forest) | 0.330~0.377 | 0.800~0.844 |
| $t_a$ | 18.08~20.08(grassland); 17.09~19.09(forest) | 0.333~0.371 | 0.783~0.841 |



**Figure 1: The geographic location, land cover, spatial distribution of precipitation and meteorological stations locations of NECT.**





Figure 2: Optimum canopy state (from Eagleson, 2002).





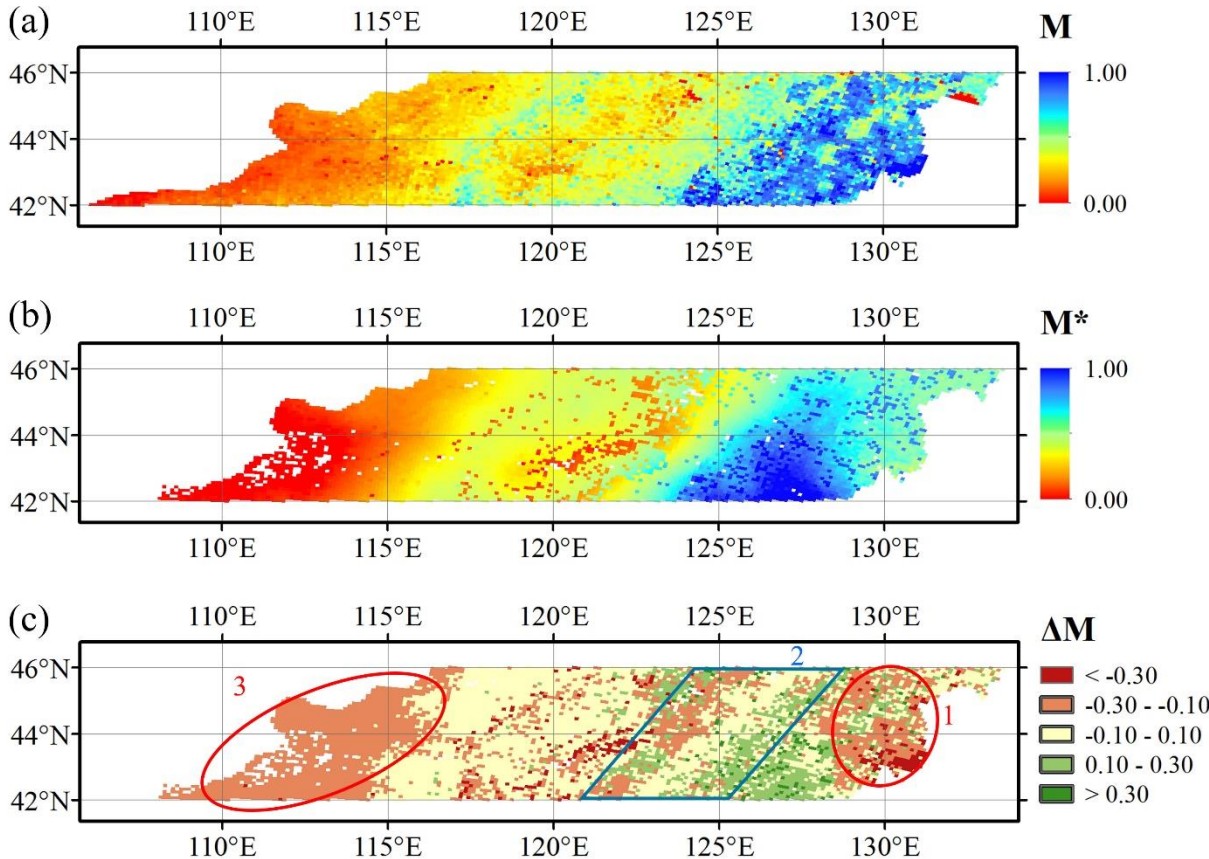

**Figure 3: Spatial distribution of mean canopy cover from MODISdata, optimal canopy cover and their differences.**



**Figure 4: Variation of *M\** , *M* during precipitation (*P$_\tau$*) and air tempereture (*t$_a$*)2000-2013 ((a) grassland; (b) forest).**





**Figure 5: *M\* *changes with *β*, *LAI* and *hₛ/h* ((a) ~ (c) grassland; (d) ~ (f) forest).**




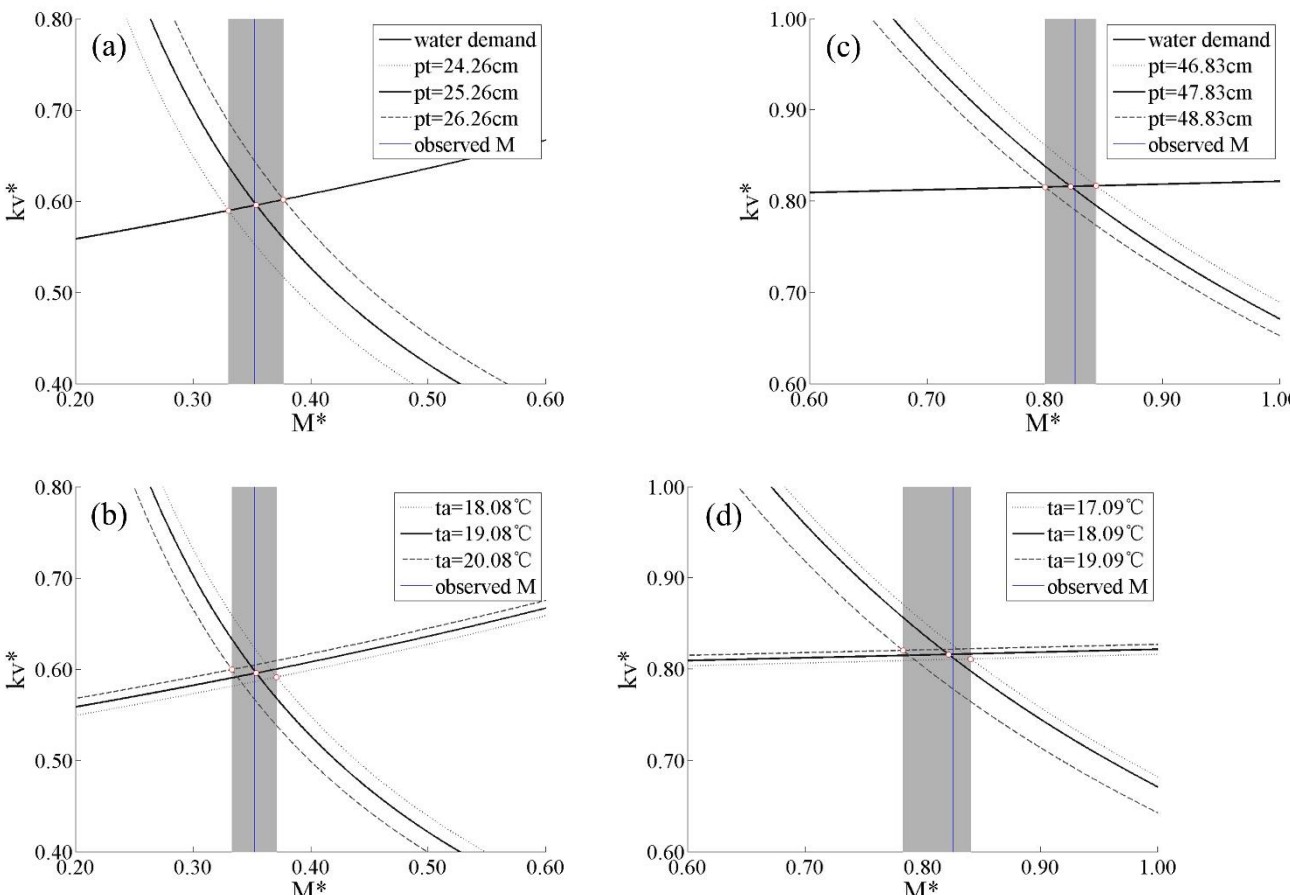

**Figure 6:** *M\* changes with precipitation (P$_\tau$) and air tempereture (t$_a$) ((a) ~ (b) grassland; (c) ~ (d) forest).*