# Peer review of "Ecohydrological Optimality in Northeast China Transect"

_Hydrology and Earth System Sciences, 2017_

## Referee Comment (RC1) · M. Coenders-Gerrits (Referee) · 17 Feb 2017

The authors present a study where they apply the Eagleson ecohydrological optimality method to a North-East transect in China. The paper is well written and structured while some language issues should be solved. Furthermore, the paper needs some clarification on the definition of some hydrological terms and the units should be checked. Since I reviewed an earlier version of this manuscript before, I don't have many comments on the scientific methodology. Nonetheless, I think, the study remains poor in its added value/novelty. Mainly it's an application of Eaglesons theory and shows a comparison between existing canopy cover versus optimal canopy cover. On the other hand, the sensitivity analysis of several climate and plant physiological parameters does provide new insights.

[Figure]

**Specific comments:**

- P1L8: Explain why "it's interesting to understand vegetation distribution..". That something is interesting is not a 'knowledge gap'. Maybe the last sentence of the abstract is the answer.
- P1L17: ".. the increas of LAI, stem fraction, AND temperature...."
- P1L22: "Transect stud**ies play** an important..."
- P2L22: too much =>many
- P2L24: ".. Eagleson present**s** a theory.." (also skip terms like 'smart')
- P2L29: ".. which IS partly due..."
- P2L32: "Mo appl**IED** this method..."
- P3L7: unit of annual rainfall is mm/year
- P3L15: "... on THE NASSA website.."
- P3L25: Why is it so difficult to define barren soil and forest? This has nothing to do with the period 2000-2013, does it?
- P3L29: Why is canopy cover set as 1 the the NDVI is above it's max value? NDVI is related to the greenness and in principle has not much to do with canopy coverage, right?
- P4L4: Rn should be in italic
- P4L5: Max and min temperature are not required for Penman (-Monteith).
- P4L10: ".. He considered THAT climate and vegetation.."
- P4L23-32: Please don't use calories as a unit. Please check official SI units (so Joule)
- P5L9: Why can the resistance ratio be fixed once the plant species is known? $r_a$ is dependent on wind and thus will change, right?
- P5L11: ".. M can also be described by THE water balance equation. In THE growing season.... of the soil column ARE:"
- P5L13:Eq 6 is not clear to me. What is meant by storm surface retention. What is $E_{T\tau}$? And how does this relate to the water balance components of Table 2? Furthermore, the E of expected value should not be in italic

- P5L26: Units don't match. ve/mtEps = [mm]/[h][mm/d]. Assuming that the unit hours should be days, the still the outcome is then days, which is not equivalent to $Mk_v$. Please verify and correct
- P5L30: remove line break.
- P6L16: ".. simulation resultS are..."
- P6L22: small=> poor
- P6L30: Would it be possible to add irrigation into Eaglesons theory? That would e.g. be an added values for this paper.
- P7L16: ".. the RESEARCH STUDIES conducted before..."
- P7L18: area => areas
- P7L22: area => areas
- P7L31: $\beta$ and $hs$ and $h$ are not explained
- P10L10-29: Some of these explanations should go in the main text. For example the definition of Er, and Et.
- P10L28: Please keep your units consistent. Here Epd has unit [mm/day] while E(Er) is in mm.
- Table 2: how is interception, runoff and evaporation calculated and what is the relation with Equation 6?
- Fig 1: unit of annual rainfall is [mm/y]
- Fig 3: Add in legend that $\Delta M = M * -M$
- Fig 5  6: lower right figure is not alligned well.

---

## Referee Comment (RC2) · Anonymous Referee #2 · 12 Mar 2017

**Review of "Ecohydrological Optimality in Northeast China Transect"**

**Manuscript ID**: hess-2017-42

**Overall Recommendation**: Moderate revision

**General Comments**:

This manuscript applied Eagleson's ecohydrological optimality method to derive the optimal canopy cover (M*) and then compared M* with the satellite derived canopy cover based on NDVI. In addition, the authors presented a comprehensive sensitivity analysis in terms of how the optimal canopy cover varies with different vegetation characteristics and climate factors. Overall, this paper is well written and organized and the main conclusions are sound. Although it is a local study focusing on the Northeast China Transect region, from a practical perspective, the implementation of Eagleson's ecohydrological optimality theory can certainly provide some insights in terms of the understanding of climate change impacts on canopy cover dynamics and therefore, can provide useful guidelines for eco-restoration projects, especially for the selection of vegetation species and plant density. However, this paper would benefit from an additional efforts of the authors to improve their writing, as there are some grammar errors and inappropriate wording (listed in the end).

Based on the above considerations, I recommend the manuscript to be returned to the authors for moderate revisions before it can be accepted.

**Specific Comments:**

1. Page 2 Line 4-5: "Vegetation is considered as the indicator of climate". This is not accurate. Please modify.
2. Page 2 Line 16-17: It is quite vague to say "Other studies … NECT". What kind of effects? Please be specific.
3. Page 2 Line 29: What does "due to the limitation of long term average state" mean?
4. For the introduction section, the importance of Eagleson's ecohydrological optimality theory should be well elaborated to better and clearly state the objective and motivation of this study.
5. Page 3 Line 6: What does "global change gradient" mean?
6. Page 3 Line 17: I do not understand why the high resolution (500 m and 1 km) datasets were resampled to coarse resolution (10 km).
7. In the methodology sections, please use SI units for variables. Please also check other places.
8. Page 6 Line 1-2: Please consider moving "This method … Table 1." to somewhere in Section 2.
9. Page 6 Line 22: What kind of "human activity"?
10. Page 7 Line 9-10: The authors mentioned that "This can be explained … climate changes." But how?

11. Page 7 Line 14-15: How are water balance components calculated? There is no description.
12. Page 7 Line 22: Please consider changing "within the observed range" to "consistent with previous studies".
13. Page 9 Line 10-16: This part could be moved to or mentioned in the introduction section to clearly state the motivation of this study.
14. Please carefully check Equations A.10 and A.12. Ve does not have the correct form.
15. For Table 1, it would be beneficial to the reader to have some statements linking it with the methods section. For example, some sentences can be added after Line 28 on Page 5.
16. Table 2, how is interception calculated?
17. In Figure 2, water supply curve should be corresponding to Eq. 7 and water demand curve should be corresponding to Eq. 5.
18. Please rephrase the caption of Figure 4.
19. Figure 5 & 6: What are the shaded areas?

**Technical Corrections (Not an Exhaustive List):**
1. Page 1 Line 6: Add "the" before "International".
2. Page 1 Line 11: Change "trade-off of" to "trade-off between".
3. Page I line 13: Change "then compare … to discuss" to "which is compared with M to further discuss".
4. Page 1 Line 15: Change "The result" to "Results", change "matches" to "match".
5. Page 1 Line 18: Change "climate change to" to "climate change on" and delete "quantitatively".
6. Page 1 Line 22: Add "." at the end of this sentence.
7. Page 1 Line 28: Change "the vegetation types" to "vegetation types".
8. Page 2 Line 3: Change "effected" to "affected".
9. Page 2 Line 7: Change "common index" to "common indexes".
10. Page 2 Line 13: Change "researches about" to "studies on".
11. Page 2 Line 25: Change "the vegetation" to "vegetation".
12. Page 2 Line 26-27: Change "light … state" to "light, energy, water and soil conditions in a long term average state.".
13. Page 2 Line 27: Add "fact that" after "Despite the".
14. Page 2 Line 28: Change "researches" to "studies". Please also check other places.
15. Page 2 Line 31: Change "trade-off of" to "trade-off between".
16. Page 2 Line 32: Change "Mo (2015)" to "Mo et al. (2015)".
17. Page 3 Line 8: Change "200mm/year" to "200 mm".
18. Page 5 Line 11: Change "be also" to "also be", add "the" before "growing".
19. Page 5 Line 12: Add "can be described as" after "soil column".
20. Page 6 Line 21: Change "is relatively" to "are relatively".
21. Page 7 Line 1: Change "effected" to "affected".
22. Page 7 Line 4: The sentence "The correlation … is 0.81" repeats previous one. Please either modify or delete this sentence.

23. Page 7 Line 5: Change "researches" to "studies".
24. Page 7 Line 6: Change "region scales" to "regions".
25. Page 8 Line 28: Change "shows the" to "shows a".
26. Page 8 Line 32: Change "to" to "from".
27. Page 9 Line 20: Change "to" to "on", change "The main … follows:" to "Main conclusions are summarized as follows:".

---

## Author Comment (AC1) · 3 Apr 2017

The authors present a study where they apply the Eagleson ecohydrological optimality method to a North-East transect in China. The paper is well written and structured while some language issues should be solved. Furthermore, the paper needs some clarification on the definition of some hydrological terms and the units should be checked. Since I reviewed an earlier version of this manuscript before, I don't have many comments on the scientific methodology. Nonetheless, I think, the study remains poor in its added value/novelty. Mainly it's an application of Eaglesons theory and shows a comparison between existing canopy cover versus optimal canopy cover. On the other hand, the sensitivity analysis of several climate and plant physiological parameters does provide new insights. RESPONSE: Thank a lot for the nice work by Dr. M. Coenders-Gerrits. We recognize that the innovation of this paper is not

remarkable but "the sensitivity analysis of several climate and plant physiological parameters does provide new insights." We had accepted all the advices about the writing and had revised them in the manuscript. SPECIFIC COMMENTS 1. P1L8: Explain why "it's interesting to understand vegetation distribution..". That something is interesting is not a 'knowledge gap'. Maybe the last sentence of the abstract is the answer. RESPONSE: It is true that something is interesting is not a 'knowledge gap'. We changed this sentence into "It is remarkable to understand vegetation distribution and dynamics under climate change in this transect." 2. P1L17: ".. the increas of LAI, stem fraction, AND temperature...." RESPONSE: It is accepted. 3. P1L22: "Transect studies play an important..." RESPONSE: It is accepted. 4. P2L22: too much =>many. RESPONSE: It is accepted. 5. P2L24: ".. Eagleson presents a theory.." (also skip terms like 'smart') RESPONSE: It is accepted. 6. P2L29: ".. which IS partly due..." RESPONSE: It is accepted. 7. P2L32: "Mo applIED this method..." RESPONSE: It is accepted. 8. P3L7: unit of annual rainfall is mm/year RESPONSE: It is accepted. 9. P3L15: "... on THE NASSA website.." RESPONSE: It is accepted. 10. P3L25: Why is it so difficult to define barren soil and forest? This has nothing to do with the period 2000-2013, does it? RESPONSE: We are sorry the confusion by the writing. We got the yearly Land Cover Types and the corresponding NDVI. Since the land cove of a fixed grid maybe changed in different year, so it is not easy to define the real barren soil or the forest areas. Our solution is that we considered the area sensed as barren soil for every year is the barren soil area, and the NDVImin is the spatial average of barren area NDVI. Similarly, the area sensed as forests every year is considered as forests area, and the spatial average of forests NDVI is NDVImax. We wrote this sentence into "Since the land cove of a fixed grid maybe changed in different year, it was hard to define the real barren soil or the forest areas." 11. P3L29: Why is canopy cover set as 1 the the NDVI is above it's max value? NDVI is related to the greenness and in principle has not much to do with canopy coverage, right? RESPONSE: According to our definition of NDVImin and NDVImax, the actual NDVI of a grid may larger than NDVImax or smaller than NDVImin. The canopy cover can

be regarded as 1 when the NDVI is above NDVImax and as 0 if the NDVI is below NDVImin. 12. P4L4: Rn should be in italic. RESPONSE: It is accepted. 13. P4L5: Max and min temperature are not required for Penman (-Monteith). RESPONSE: Yes, Max and min temperature are not required for Penman (-Monteith) in general. Here we used the FAO-Penman-Monteith Equation, where Max and min temperature were used to calculate the actual water vapor pressure. 14. P4L10: ".. He considered THAT climate and vegetation.." RESPONSE: It is accepted. 15. P10L28: Please keep your units consistent. Here Epd has unit [mm/day] while E(Er) is in mm. RESPONSE: All the units of evaporation are mm/day, but Er is the storm surface retention depth of each storm. So the unit of Er is mm. 16. Table 2: how is interception, runoff and evaporation calculated and what is the relation with Equation 6? RESPONSE: Yes, all the components of water balance are estimated based on Equation 6, which had been stated when the Table 2 was referred. 17. Fig 1: unit of annual rainfall is [mm/y] RESPONSE: It is accepted. 18. Fig 3: Add in legend that $\Delta M = M^* - M$. RESPONSE: It is accepted. 19. Fig 5 6: lower right figure is not alligned well. RESPONSE: The aligning problem is caused by the WORD and we had improved it.

Please also note the supplement to this comment: http://www.hydrol-earth-syst-sci-discuss.net/hess-2017-42/hess-2017-42-AC1-supplement.pdf

**Fig. 1.**

Meteorological stations — Mean annual precipitation (mm/year)

**Land cover types**

| | | |
|---|---|---|
| Water | Closed shrublands | Croplands |
| Evergreen needleleaf forest | Open shrublands | Urban and built-up |
| Evergreen broadleaf forest | Woody savannas | Cropland/Natural vegetation mosaic |
| Deciduous needleleaf forest | Savannas | Snow and ice |
| Deciduous broadleaf forest | Grasslands | Barren or sparsely vegetated |
| Mixed forest | Permanent wetlands | Fill value |

(a)

M

(b)

M*

(c)

$\Delta M = M^* - M$

| | |
|---|---|
| ■ | < -0.30 |
| ■ | -0.30 – -0.10 |
| □ | -0.10 – 0.10 |
| ■ | 0.10 – 0.30 |
| ■ | > 0.30 |

**Fig. 2.**

[Figure]

[Figure]

[Figure]

[Figure]

[Figure]

[Figure]

**Fig. 3.**

[Figure]

[Figure]

[Figure]

[Figure]

**Fig. 4.**

---

## Author Comment (AC2) · 3 Apr 2017

This manuscript applied Eagleson's ecohydrological optimality method to derive the optimal canopy cover (M*) and then compared M* with the satellite derived canopy cover based on NDVI. In addition, the authors presented a comprehensive sensitivity analysis in terms of how the optimal canopy cover varies with different vegetation characteristics and climate factors. Overall, this paper is well written and organized and the main conclusions are sound. Although it is a local study focusing on the Northeast China Transect region, from a practical perspective, the implementation of Eagleson's ecohydrological optimality theory can certainly provide some insights in terms of the understanding of climate change impacts on canopy cover dynamics and therefore, can provide useful guidelines for eco-restoration projects, especially for the selection of vegetation species and plant density. However, this paper would

benefit from an additional efforts of the authors to improve their writing, as there are some grammar errors and inappropriate wording (listed in the end). Based on the above considerations, I recommend the manuscript to be returned to the authors for moderate revisions before it can be accepted. RESPONSE: Thanks for the comments. We try to improve our writing according to the review, including grammar errors and inappropriate wording. 1. Page 2 Line 4-5: "Vegetation is considered as the indicator of climate". This is not accurate. Please modify. RESPONSE: This sentence is unnecessary and not clear here. We deleted it. 2. Page 2 Line 16-17: It is quite vague to say "Other studies ... NECT". What kind of effects? Please be specific. RESPONSE: We wrote this sentence into: "NDVI driven by climate changes varied differently between vegetation types and seasons(Piao et al. 2006). Duan et al. (2011) illustrated that precipitation was the most importance factor in affecting the temporal NDVI patterns over semi-arid and arid regions of China. Peng et al. (2012) found > 70% of the temporal variations in NDVI were contributed by precipitation during the growing season in typical and desert steppes in Northeast China. Mao et al.(2012), however, discovered that the correlation between NDVI and temperature was higher than with precipitation over most parts of Northeast China for all vegetation covers; NDVI presented a downward trend with increased temperature and remarkably decreased precipitation. Further, Yuan et al. (2015) suggested diverse responses of grasslands to precipitation intensities." 3. Page 2 Line 29: What does "due to the limitation of long term average state" mean? RESPONSE: The limitation of long term average state means the Eagleson's method can not be applied in year scale or other smaller scale, so it can not be used to solve practice problems such as water resources and ecological restoration projects. We change this sentence into "...which is partly due to the limitation of long term temporal scale..". 4. For the introduction section, the importance of Eagleson's ecohydrological optimality theory should be well elaborated to better and clearly state the objective and motivation of this study. RESPONSE: Thanks for the advice. We add some sentences in the introduction section: "The NDVI data offer us a method to estimate actual canopy cover. If we

can verified Eagleson's ecohydrological optimality theory by comparing the optimal canopy cover and remote sensing canopy cover, we can discuss the impact of climate factors and vegetation properties on vegetation cover. From this framework, we can certainly provide some insights in terms of the understanding of climate change impacts on canopy cover dynamics and therefore, can provide useful guidelines for eco-restoration projects, especially for the selection of vegetation species and plant density." 5. Page 3 Line 6: What does "global change gradient" mean? RESPONSE: It is a mistake here, not "global change gradient", just "change gradient". 6. Page 3 Line 17: I do not understand why the high resolution (500 m and 1 km) datasets were resampled to coarse resolution (10 km). RESPONSE: The spatial resolution is determined depend on the calculated amount. Since the coarse resolution does not influence the method and conclusion, the resolution of 10km x 10km is acceptable. 7. In the methodology sections, please use SI units for variables. Please also check other places. RESPONSE: We changed "597.3cal/g" into "2500J/g", "cal/(mm2Åůday)" into "J/(mm2Åůday)" , "cal/(gÅůK)" into "J/(gÅůK)". The other units are retained. 8. Page 6 Line 1-2: Please consider moving "This method . . . Table 1." to somewhere in Section 2. RESPONSE: The variables in Table 1 are introduced in Section 3, so it is strange to list them in Section 2. Therefore, we still keep Table 1 here. 9. Page 6 Line 22: What kind of "human activity"? RESPONSE: Here "human activity" refers to the Natural Forest Protection Project (NFPP), which is given in the next sentences. We changed "human activity" into "forest protection project". 10. Page 7 Line 9-10: The authors mentioned that "This can be explained . . . climate changes." But how? RESPONSE: For example, the canopy cover might not increase immediately with the increasing precipitation but might increase in next year. We add this sentence after "This can be explained . . . climate changes." 11. Page 7 Line 14-15: How are water balance components calculated? There is no description. RESPONSE: The water balance components were calculated based on Equation 6, more detail can be found in Appendix. We change this sentence into: "The proportions of the water balance components for annual average growing season are calculated for each part based

on Equation 6, as shown in Table 2." 12. Page 7 Line 22: Please consider changing "within the observed range" to "consistent with previous studies". RESPONSE: It is accepted. 13. Page 9 Line 10-16: This part could be moved to or mentioned in the introduction section to clearly state the motivation of this study. RESPONSE: We deleted this paragraph here and moved some sentences into the introduction section. 14. Please carefully check Equations A.10 and A.12. Ve does not have the correct form. RESPONSE: We are sorry for the confusion. We rewrote it into: "$\beta v$ is equal to 1.0 when the water condition reaches optimal state. When the bare soil evaporation is ignored, Eq. (A.10) can be simplified into" 15. For Table 1, it would be beneficial to the reader to have some statements linking it with the methods section. For example, some sentences can be added after Line 28 on Page 5. RESPONSE: We added some sentences in the end of Section 3. "The input data and parameters include remote sensing data, meteorological data, vegetation data and soil data. The remote sensing data are the vegetation cover and LAI. The main meteorological data are length of growing season, potential evaporation, air temperature and storm duration. The main vegetation data are surface retention depth, leaf angle and stem height. The main soil data are soil porosity and hydraulic conductivities. " 16. Table 2, how is interception calculated? RESPONSE: The interception was calculated based on Equation A.3. 17. In Figure 2, water supply curve should be corresponding to Eq. 7 and water demand curve should be corresponding to Eq. 5. RESPONSE: Yes, it is a mistake. Eq. 3 should be Eq. 5 and Eq. 5 should be Eq. 7. 18. Please rephrase the caption of Figure 4. RESPONSE: We revised it into "Variation of M* , M, precipitation (pt) and air temperature (ta) during 2000-2013 ((a) grassland; (b) forest)." 19. Figure 5 & 6: What are the shaded areas? RESPONSE: The shaded areas mean the range of M* with the change of climate factors or vegetation properties. We added it into the caption of Figure 5 and Figure 6. Technical Corrections RESPONSE: we accept most of technical corrections and thank for the careful review. 1. Page 1 Line 6: Add "the" before "International". 2. Page 1 Line 11: Change "trade-off of" to "trade-off between". 3. Page I line 13: Change "then compare . . . to discuss" to "which is compared with

M to further discuss". 4. Page 1 Line 15: Change "The result" to "Results", change "matches" to "match". 5. Page 1 Line 18: Change "climate change to" to "climate change on" and delete "quantitatively". 6. Page 1 Line 22: Add "." at the end of this sentence. 7. Page 1 Line 28: Change "the vegetation types" to "vegetation types". 8. Page 2 Line 3: Change "effected" to "affected". 9. Page 2 Line 7: Change "common index" to "common indexes". 10. Page 2 Line 13: Change "researches about" to "studies on". 11. Page 2 Line 25: Change "the vegetation" to "vegetation". 12. Page 2 Line 26-27: Change "light . . . state" to "light, energy, water and soil conditions in a long term average state.". 13. Page 2 Line 27: Add "fact that" after "Despite the". 14. Page 2 Line 28: Change "researches" to "studies". Please also check other places. 15. Page 2 Line 31: Change "trade-off of" to "trade-off between". 16. Page 2 Line 32: Change "Mo (2015)" to "Mo et al. (2015)". 17. Page 3 Line 8: Change "200mm/year" to "200 mm". 18. Page 5 Line 11: Change "be also" to "also be", add "the" before "growing". 19. Page 5 Line 12: Add "can be described as" after "soil column". 20. Page 6 Line 21: Change "is relatively" to "are relatively". 21. Page 7 Line 1: Change "effected" to "affected". 22. Page 7 Line 4: The sentence "The correlation . . . is 0.81" repeats previous one. Please either modify or delete this sentence. 23. Page 7 Line 5: Change "researches" to "studies". 24. Page 7 Line 6: Change "region scales" to "regions". 25. Page 8 Line 28: Change "shows the" to "shows a". 26. Page 8 Line 32: Change "to" to "from". 27. Page 9 Line 20: Change "to" to "on", change "The main . . . follows:" to "Main conclusions are summarized as follows:".

Please also note the supplement to this comment:
http://www.hydrol-earth-syst-sci-discuss.net/hess-2017-42/hess-2017-42-AC2-supplement.pdf

[Figure]

Fig. 1.

---

## Author Response (AR2)

RESPONSES to Reviewer 2: Anonymous Referee

I only have a few minor comments:

1. For my first round review, I understand that the variables in Table 1 are introduced in Section 3, but I was asking the authors to move the SENTENCE "This method ... Table 1." to somewhere in Section 2. Not the table itself.

RESPONSE: We are sorry to misunderstand this comment in the first round review. It really is a problem to describe the input data in the methodology section. We move these sentences in the end of Section 3 to the end of Section 2 according to the advice by Reviewer 2.

2. In the appendix, even though the author ignores the bare soil evaporation and assumes that beta_v=1, the last term in Equation (A.12) still does not have the correct form. Please carefully check the equations.

RESPONSE: We are sorry for the mistake in the equation expression. Equation A.1 is not $P_\tau = m_v m_h$ but $P_\tau = m_v/m_h$, and Equation A.12 is $
[revised manuscript text omitted]

[Figure]

**Figure 4: Variation of *M\** , *M* precipitation (*Pᵣ*) and air temperature (*tₐ*) during 2000-2013 ((a) grassland; (b) forest).**

[Figure]

**Figure 5:** *M\* changes with β, LAI and hs/h ((a) ~ (c) grassland; (d) ~ (f) forest). The shaded areas mean the range of M\* with the change of climate factors.*

[Figure]

**Figure 6:** *M\** **changes with precipitation (*$P_\tau$*) and air temperature (*$t_a$*) ((a) ~ (b) grassland; (c) ~ (d) forest). The shaded areas mean the range of M\* with the change of vegetation properties.**